# CPPs to the Test: Effects on Binding, Uptake and Biodistribution of a Tumor Targeting Nanobody

**DOI:** 10.3390/ph14070602

**Published:** 2021-06-23

**Authors:** Estel Collado Camps, Sanne A. M. van Lith, Cathelijne Frielink, Jordi Lankhof, Ingrid Dijkgraaf, Martin Gotthardt, Roland Brock

**Affiliations:** 1Department of Biochemistry, Radboud Institute for Molecular Life Sciences, Radboudumc, 6525 GA Nijmegen, The Netherlands; e.colladocamps@radboudumc.nl (E.C.C.); jordi.lankhof@radboudumc.nl (J.L.); 2Department of Medical Imaging, Nuclear Medicine, Radboudumc, 6525 GA Nijmegen, The Netherlands; sanne.vanlith@radboudumc.nl (S.A.M.v.L.); cathelijne.frielink@radboudumc.nl (C.F.); martin.gotthardt@radboudumc.nl (M.G.); 3Department of Biochemistry, Maastricht University, 6229 ER Maastricht, The Netherlands; i.dijkgraaf@maastrichtuniversity.nl; 4Department of Medical Biochemistry, College of Medicine and Medical Sciences, Arabian Gulf University, Manama 293, Bahrain

**Keywords:** cell-penetrating peptides, molecular imaging, nanobody, spheroids, tumor targeting, biodistribution

## Abstract

Nanobodies are well-established targeting ligands for molecular imaging and therapy. Their short circulation time enables early imaging and reduces systemic radiation exposure. However, shorter circulation time leads to lower tracer accumulation in the target tissue. Cell-penetrating peptides (CPPs) improve cellular uptake of various cargoes, including nanobodies. CPPs could enhance tissue retention without compromising rapid clearance. However, systematic investigations on how the functionalities of nanobody and CPP combine with each other at the level of 2D and 3D cell cultures and in vivo are lacking. Here, we demonstrate that conjugates of the epidermal growth factor receptor (EGFR)-binding nanobody 7D12 with different CPPs (nonaarginine, penetratin, Tat and hLF) differ with respect to cell binding and induction of endocytosis. For nonaarginine and penetratin we compared the competition of EGF binding and performance of L- and D-peptide stereoisomers, and tested the D-peptide conjugates in tumor cell spheroids and in vivo. The D-peptide conjugates showed better penetration into spheroids than the unconjugated 7D12. Both in vivo and in vitro, the behavior of the agent reflects the combination of both functionalities. Although CPPs cause promising increases in in vitro uptake and 3D penetration, the dominant effect of the CPP in the control of biodistribution warrants further investigation.

## 1. Introduction

Nanobodies are soluble antigen binding proteins derived from the variable part of the single heavy chain of camelid antibodies (VHH) [1]. They are approximately ten times smaller than a full monoclonal antibody (about 15 kDa), very stable and easy to produce. Furthermore, they can be generated and selected to bind a wide range of epitopes, with affinities in the nanomolar range. Nanobodies can also recognize concave epitopes, which are structurally inaccessible for antibodies [2,3]. Due to their small size, they clear faster from the circulation than antibodies. In addition, in vitro and in vivo results demonstrate that they penetrate deeper into tissue, reaching a more homogeneous distribution throughout the whole tumor mass [4,5]. These characteristics make nanobodies excellent tools for molecular imaging, targeted therapy and the combination thereof, named theranostics. Their potential as imaging and/or therapeutic tracers has already been shown for a number of targets [6].

However, there are challenges to overcome. Fast blood clearance provides good contrast at early timepoints, but narrows the time window in which a nanobody can reach the target tissue. For optimal imaging, clearance and tissue uptake should be in balance. In preclinical tumor models, specific tumor accumulation of nanobodies is typically ≤10%ID/g at the optimal timepoint [7,8,9], compared to the 30%ID/g or higher of some monoclonal antibodies [5,10,11]. Several approaches that increase tissue targeting also increase the molecular weight, shifting the balance back to increased plasma half-lives, and possibly compromising tissue penetration. One example is the development of nanobody-based multispecific or multivalent tracers [12,13]. Other modifications that increase the plasma half-life are addition of albumin-binding domains and PEGylation [14,15,16,17,18].

Ideally, rapid and specific tissue uptake should be enhanced without increasing the molecular weight, to preserve fast clearance of an unbound nanobody. One strategy is to engineer the nanobody sequence itself, by substituting amino acids at certain positions of the original sequence to add positive charges to the surface [17]. This approach has also been applied to antibodies [18]. However, this “cationization” or “re-surfacing” increases retention in all organs and can also reduce affinity, counteracting specific targeting and augmenting radiation exposure of healthy tissues.

In order to enhance specificity, addition of tumor targeting peptides has also been explored. These peptides are only a few amino acids long and thus do not considerably change the molecular weight of the targeting agent. In this direction the addition of an RGD or iRGD peptide, which targets integrin αvβ3, has attracted considerable attention [19,20,21]. iRGD addition has also been pursued for a nanobody and an antibody [19,22]. CendR peptides, or cleavable forms of them, are other examples. The CendR motif binds to Neuropilin-1, and this interaction helps endocytosis [23]. However, integrins and Neuropilin-1 are also expressed in healthy tissues.

Cell-penetrating peptides (CPPs) induce cellular uptake in a receptor-independent manner. Since their discovery in the 90s [24,25] they have been employed to mediate uptake of oligonucleotides [26], peptides [27], antibodies [28,29] and nanoparticles [30,31]. Therefore, CPPs are also good candidates to enhance cellular uptake and tissue retention of tumor targeting agents.

Previously, we conjugated the anti-EGFR nanobody 7D12 to a CPP derived from human lactoferrin (hLF) [32]. This conjugate was taken up through endocytosis to a higher extent than the unconjugated 7D12 [33]. This uptake was independent of EGFR activation, demonstrating synergy between receptor targeting by the nanobody and internalization triggered by the CPP. Moreover, we recently found that CPPs rapidly penetrate into tumor cell spheroids [34]. The capacity to penetrate was more pronounced for the CPP nonaarginine (R9) than for the amphipathic penetratin (Pen) peptide and D-peptides were superior to their L-peptide counterparts. Furthermore, R9 enhanced spheroid penetration of elastin-like polypeptide nanoparticles [30].

Both observations combined suggest that nanobody-CPP conjugates have the potential to specifically target, penetrate and be retained in tumors in vivo. However, to exploit this potential fully, more mechanistic insight is needed. As CPPs differ in terms of charge, secondary structure and amphiphilicity, it is plausible that also the respective nanobody-CPP conjugates will interact differently with cells and tissues. In the context of theranostics, it will be of particular interest to understand to which extent the targeting functionality can counteract the propensity of the CPP to distribute to the liver and kidneys [35,36]. To understand the interplay between nanobody (or targeting moiety) and CPP, we followed up on our previous results by conjugating 7D12 to four different CPPs: hLF, R9, Tat and Pen. R9, Tat and hLF are cationic CPPs that all show comparable cellular uptake characteristics [37], while Pen is more amphipathic.

We assessed binding and uptake of the conjugates on cell monolayers, also in the presence of the natural EGFR ligand EGF. Furthermore, for the penetratin and nonaarginine conjugates (7D12-Pen and 7D12-R9), we probed EGFR phosphorylation, and binding and uptake at low concentrations of radiolabeled conjugate. We included conjugates of the D-peptide variants in the comparison, and evaluated their 3D penetration and retention in cell spheroids and in vivo biodistribution in mouse models. Each CPP enhanced binding and internalization of 7D12 in a distinct manner. For the conjugates tested, CPPs enhanced retention and penetration into SKOV3 spheroids. However, this did not translate into higher tumor uptake in an in vivo model.

## 2. Results

### 2.1. Production of Nanobody-CPP Conjugates

The recombinantly expressed and purified 7D12 anti-EGFR nanobody was coupled to either the fluorophore Atto532 or the chelator DTPA via maleimide chemistry (Figure 1A), and to each CPP via Sortase A-mediated transpeptidation (Figure 1B, CPP sequences in Appendix A). All conjugates were pure and were in the expected size range as confirmed by SDS-PAGE stained with Coomassie brilliant blue (Figure 1B,C) and ESI-ToF mass spectrometry (Appendix A). Observed masses were very close to the calculated masses. 

### 2.2. Differential Uptake of 7D12-CPP Conjugates in EGFR Expressing Cells

First, we evaluated binding and uptake of the 7D12-Atto532-CPP conjugates in A431 cell monolayers, using live-cell confocal microscopy. Conjugates were used in low micromolar concentrations as reported previously [33]. 

All nanobody variants showed endosomal uptake (Figure 2A). Visually, the presence of endosomes was higher for 7D12-R9, 7D12-Pen and 7D12-Tat than for 7D12-hLF and unconjugated 7D12. Membrane localization was less intense for 7D12-Atto532-Pen and 7D12-Atto-532-hLF than for the other variants. Visual inspection suggested more vesicular uptake in relation to the membrane-bound fluorescence for 7D12-Atto532-Pen. Total fluorescence intensity (total cell binding) was significantly lower for 7D12-Atto532-Pen, 7D12-Atto532-Tat and 7D12-Atto532-hLF in comparison to 7D12-Atto532. Signal intensities of 7D12-Atto532-R9 and 7D12-Atto532 were similar (Figure 2B).

Of the CPPs we tested, nonaarginine and penetratin differ the most in their amphipathicity. Although penetratin showed an overall lower intensity and lower cell membrane binding, the low membrane staining and high number of endosomes suggests that its internalization was more efficient in respect to the amount of binding. In previous research, we had observed that nonaarginine showed a faster and deeper penetration of tumor cell spheroids than penetratin [34]. As our goal was the application of the conjugates in vivo, for a further comparison of conjugates we selected these two peptides.

### 2.3. Competition of EGF Binding by 7D12-R9 and 7D12-Pen

The binding site of 7D12 on EGFR overlaps with that of the natural ligand EGF [38]. Therefore, we were interested on whether unconjugated 7D12 and the 7D12-CPP conjugates differed in the competition with EGF. To avoid internalization of EGFR, A431 cells were incubated with the compounds at 4 °C.

In the “coincubation” experiments, cells were preincubated with nanobody or conjugate and EGF was added in the presence of conjugate after 30 min. 7D12-Atto532-CPP conjugates were slightly but consistently better in competing with EGF binding compared to 7D12-Atto532 (Figure 3A). However, none of the conjugates showed better binding than unconjugated 7D12 in coincubation with EGF (Figure 3B). EGF has a low nanomolar affinity for EGFR while the KD of 7D12 is in the upper nanomolar range [33,38,39]. Competition with EGF may explain the smaller difference between 7D12-Pen and 7D12 in this experiment (Figure 3B), compared with the previous one (Figure 2).

In the “sequential incubation” experiments, cells were first incubated with 7D12 or 7D12-CPP and subsequently with EGF, in the absence of 7D12 or conjugate. Again, both conjugates were more effective in competing for EGF compared to unconjugated 7D12-Atto532 (Figure 3C). Notably, 7D12-Atto532-R9 led to a significantly higher reduction of EGF binding than 7D12-Atto532-Pen. Additionally, both conjugates showed stronger binding than 7D12-Atto532 in these conditions (Figure 3D). These results are in line with a stronger membrane binding of 7D12-R9 compared to 7D12-Pen.

### 2.4. The Effect of 7D12-R9 and 7D12-Pen on EGFR Activation

In our previous work we showed that 7D12-hLF induces receptor internalization without activation. We assessed whether this also holds for 7D12-Atto532-R9 and 7D12-Atto532-Pen. A431 cells were incubated with 7D12-Atto532 and each CPP-conjugate in low serum conditions. EGF was added for the last 10 min before fixating the cells, and EGFR phosphorylation was detected by immunofluorescence. 

The conjugates reduced EGFR phosphorylation to the same extent as the unconjugated nanobody. Compared to 7D12-Atto532, both CPP-conjugates caused a small increase in phosphorylation in the absence of EGF stimulation (Figure 4A), however the difference was not significant. As observed in the sequential incubation experiment at 4 °C (Figure 3D), the amount of bound 7D12-Atto532-R9 conjugate was significantly higher than that of 7D12-Atto352 and 7D12-Atto532-Pen (Figure 4B).

### 2.5. Binding and Uptake of 7D12-DTPA-R9 and 7D12-DTPA-Pen and Their D-Peptide Counterparts at Picomolar Concentrations

The proteolytic stability of D-amino acid peptides is advantageous for in vivo applications. However, we previously observed lower endosomal uptake of all D-amino acid CPPs compared to their L-amino acid counterparts [40]. For nanobody conjugates, confocal microscopy revealed little visual difference between L- and D-peptide pairs (Appendix A). Nevertheless, D-peptide conjugates did show lower total fluorescence (Appendix A). Furthermore, in sequential EGF incubation (as explained for Figure 3), both L- and D-peptide conjugates inhibited EGF binding to a similar degree (Appendix A). In view of these results, and in preparation of in vivo experiments, we assessed binding and internalization of radiolabeled conjugates of both L- and D-peptides.

Cells were incubated with [^111^In]In-DTPA-7D12, [^111^In]In-DTPA-7D12-R9 and [^111^In]In-DTPA-7D12-Pen and their D-amino acid counterparts at an approximate concentration of 5 pM (Figure 5). The conjugates were radiolabeled with a specific activity 0.37 MBq/µg. Purity was checked by iTLC and exceeded 90%. Interestingly, binding and internalization differed strongly between CPPs and between the stereoisomers of one CPP.

Conjugated [^111^In]In-DTPA-7D12 showed good binding (28.0% ± 0.2%), and internalization (6.0% ± 0.1%) in A431 cells after 1 h incubation (Figure 5A). Both binding and internalization were completely blocked with excess of non-radiolabeled unconjugated 7D12, and there was minimal binding to negative control HEK293 cells (1.2% ± 0.04%), confirming EGFR specificity (Figure 5B). Binding of [^111^In]In-DTPA-7D12-R9 to A431 cells was notably higher compared to [^111^In]In-DTPA-7D12 (39.0% ± 0.7% vs. 28.0% ± 0.2%), while internalization was not remarkably altered (7.5% ± 0.4% vs. 6.0% ± 0.1%). In contrast to the unconjugated nanobody, binding and internalization of [^111^In]In-DTPA-7D12-R9 could not be blocked. Additionally, binding to and internalization in HEK293 cells was observed (33.0% ± 2.8% and 4.8% ± 0.3%, respectively) demonstrating a prominent role of the CPP in the cell association. 

The binding pattern of the penetratin conjugate differed considerably from that of the R9 conjugate. [^111^In]In-DTPA-7D12-Pen showed significantly increased binding (39.0% ± 0.7% vs. 28.0% ± 0.2%) and internalization (17% ± 1.3% vs. 6.0% ± 0.1%) in A431 cells when compared to [^111^In]In-DTPA-7D12. However, in contrast to the R9 conjugate, binding could partly be blocked by unlabeled 7D12 (20% ± 0.5% binding and 12% ± 0.2% internalization when blocked). Increased binding to and internalization into HEK293 cells was also observed for [^111^In]In-DTPA-7D12-Pen (12.5% ± 1.3% and 5.7% ± 1.6%, respectively), however, it was notably lower than for [^111^In]In-DTPA-7D12-R9. Remarkably, binding of [^111^In]In-DTPA-7D12-Pen to HEK cells (EGFR negative) could also partly be blocked by unlabeled 7D12, which suggests that the nanobody contributes to some unspecific interactions with the cell surface.

For the D-peptide conjugates, binding to A431 cells could be blocked to a larger extent than for the L-peptides. On HEK cells, binding was less. For both cell lines, the D-peptides internalized less than the L-peptides. Overall, these results demonstrate weaker binding of the D-peptide conjugates, which renders the interaction more nanobody dependent. 

### 2.6. D Tissue Penetration and Retention in Cell Spheroids

Next, we addressed the penetration of the conjugates into cell spheroids. We focused on 7D12-r9 and 7D12-pen (the D-peptide conjugates) as r9 and pen had outperformed their L-analogues in previous 3D experiments [34], most likely due to their enhanced proteolytic stability. 

Next to the A431 cells, which highly overexpress EGFR, we also used SKOV3 cells with a medium-high EGFR expression (Figure 6). Monolayers of both cell-lines were tested in parallel for comparison with the 3D observations (Appendix A). An anti-GFP nanobody was included as a control to single out the impact of the CPP. For this nanobody, conjugation with penetratin was repeatedly unsuccessful. Therefore, only results for the anti-GFP-Atto532-r9 conjugate are shown.

The 7D12-Atto532-r9 conjugate and anti-GFP-Atto532-r9 fully penetrated the SKOV3 cell spheroids. 7D12-Atto532-pen reached some cells in the core of the spheroid but showed a brighter signal at the periphery. The unconjugated nanobodies showed almost no retention in the spheroids, demonstrating that in the SKOV3 spheroids 3D penetration is mostly peptide mediated (Figure 6A). 

In contrast, for the A431 spheroids penetration was incomplete, with all compounds showing strong binding to the rim (Figure 6B). However, also for the anti-GFP-Atto532-r9 conjugate only very limited penetration was observed which can best be explained by a higher density of the A431 spheroids in comparison to the SKOV3 cell spheroids. Therefore, for the EGFR binding conjugates, the restriction of binding to the rim may also be a function of cell density rather than the high density of EGFR that may act as a binding site barrier [41]. Next to the limited penetration of the non-targeted anti-GFP-r9 conjugate also the fact that A431 spheroids were smaller in spite of the same cell number supports this explanation. 

In SKOV3 cell monolayers, uptake of both conjugates was higher compared to unconjugated 7D12 (Appendix A). In A431 cell monolayers unconjugated 7D12 showed strong membrane staining. The conjugates showed lower fluorescence intensity but more endosomal uptake, and there were no remarkable differences between them (Appendix A). In all conditions, the anti-GFP-Atto532-r9 conjugate showed CPP-driven internalization, notably without membrane staining on cell monolayers. No uptake was observed for the unconjugated anti-GFP nanobody. 

Notably, these experiments were performed at a concentration of 0.5 µM instead of 2 µM, because it was technically not possible to reach higher concentrations in the small culture volumes of the spheroids. This could account for the smaller differences between 7D12-Atto532-r9 and 7D12-Atto532-pen, in comparison to the previous experiments.

Both in 2D and 3D, total fluorescence intensity was lower in SKOV3 cells than in A431 cells, likely reflecting the lower receptor expression. Furthermore, the differences between 7D12-Atto532-r9 and 7D12-Atto532-pen were clearer in SKOV3 cells than in A431 cells. This suggests that when nanobody-driven interactions were less prominent (due to lower receptor expression) the role of the CPP gains importance. In this context, penetratin was less active as a CPP than nonaarginine.

### 2.7. Biodistribution and SPECT Imaging of the r9 and Pen Conjugates In Vivo

Finally, we assessed the in vivo biodistribution of the D-peptide conjugates in nude mice. To compare accumulation in tumors with different EGFR expression, mice were inoculated with an A431 tumor on the left flank and a SKOV3 tumor on the right flank. Tracer accumulation in tissues was measured at 4 h and 24 h post-injection. The complete summary of the results can be found in Appendix A.

Overall, tracer accumulation was higher in A431 than in SKOV3 tumors (9.6% ± 2.3% vs. 2.2% ± 0.7% for 7D12, 2.0% ± 0.4% vs. 1.1% ± 0.0% for 7D12-r9 and 5.0% ± 0.3% vs. 2.0% ± 0.5% for 7D12-pen at 4 h post-injection), and it was higher at 4 h than at 24 h post-injection (4.8% ± 0.4% for 7D12, 1.2% ± 0.2% for 7D12-r9 and 2.7% ± 0.7% for 7D12-pen in A431 tumors). Unconjugated [111In]In-DTPA-7D12 showed the highest tumor uptake (Figure 7). Bigger tumors accumulated more tracer in absolute values, but the variability in uptake did not depend on tumor size when uptake was corrected for organ weight (Appendix A). [111In]In-DTPA-7D12-pen showed higher tumor accumulation than [111In]In-DTPA-7D12-r9. In line with the results of the binding and uptake assays, tumor accumulation of [111In]In-DTPA-7D12-pen was more efficiently blocked than that of [111In]In-DTPA-7D12-r9. We observed an increase in liver uptake for both conjugates, while kidney uptake was lower, in comparison to unconjugated 7D12. 

Tumor accumulation was too low and kidney and liver uptake too high to be able to visualize the tumors on SPECT/CT. Only for [111In]In-DTPA-7D12 we could visualize a faint signal at the A431 tumor region at 4 h post-injection (Figure 8).

## 3. Discussion

The goal of this study was to gain a thorough understanding of how different CPPs affect the interaction of a nanobody with cells and tissues. For this purpose, we conjugated the well-characterized anti-EGFR nanobody 7D12 to a set of well-established CPPs. We also addressed the impact of the stereochemistry of two of the CPPs. Importantly, we combined semiquantitative imaging and quantitative radioactivity-based techniques and went from simple in vitro monolayer models to in vivo models. 

In comparison to the unconjugated nanobody, only 7D12-R9 showed the same overall intensity in fluorescence microscopy. 7D12-Pen, 7D12-Tat and 7D12-hLF showed lower total intensity, although visual examination showed high endocytosis of 7D12-Pen and 7D12-Tat. 7D12-R9 showed very good cell association, and 7D12-Pen showed high endocytosis even with lower membrane staining. These results led to the hypothesis that penetratin induces rapid internalization while nonaarginine induces more long-lasting membrane interaction.

Notably, the results in the subsequent experiments were consistent with this hypothesis. In binding assays at low concentrations, an excess of unlabeled nanobody blocked the binding of penetratin conjugates more strongly than that of the nonaarginine conjugates, but penetratin conjugates had a stronger capacity to internalize. The results of co-incubation experiments with EGF were along the same line. Higher amounts of R9 remained bound to the cells, thereby blocking EGF binding more effectively than unconjugated 7D12 and 7D12-Pen (Figure 3). Both nonaarginine and penetratin are cationic CPPs, but the former is arginine rich while the latter is amphipathic [42]. For tetrameric CPP-conjugated antibodies, highly cationic CPPs triggered slightly more uptake than amphipathic ones [28], while the 7D12-nonaarginine showed more surface binding, but not more uptake, compared with the 7D12-penetratin ones.

In spite of the higher internalization and slightly better EGF competition, 7D12-R9 and 7D12-Pen were not better than the unconjugated 7D12 in reducing EGFR activation. This is in contrast with the results obtained for the 7D12-hLF conjugate [33]. Although it would have been of added value, reduction of EGFR activation is not a requisite for the improvement of theranostic compounds, which is the long-term goal of the work presented here.

The protease resistance of D-peptides can be advantageous for in vivo applications [43]. Therefore, we compared the conjugates of the D- and L-amino acid CPPs. Confocal microscopy and radioactivity-based assays revealed similar uptake and binding patterns for D- and L-conjugates, albeit total intensity and retained percentage were lower for the D-peptide conjugates. Importantly, lower cellular uptake of the D-peptide variants had been observed before for unconjugated CPPs [40]. This provides evidence that the L- and D-CPPs differ in their interactions with the cell surface. However, the molecular basis for these differences is not clear. Further research should elucidate whether these differences relate to the glycocalyx, the lipid bilayer or cell surface proteins. In spite of the lower cell interaction, given protease resistance, similar overall behavior for the conjugate and good 3D penetration, we considered that the D-peptide conjugates were preferable for experiments in vivo.

By comparing binding and uptake in A431 cells with those on SKOV3 cells, which express EGFR to different levels, we gained information on the relative contribution of nanobody-mediated receptor binding and CPP-mediated interactions. On SKOV3 monolayers 7D12-r9 showed clear membrane staining. In contrast, 7D12-pen was almost only present in vesicular structures, pointing to a much weaker membrane binding. On A431 monolayers both conjugates were found on the membrane and in vesicles. This observation, and the comparison to the anti-GFP conjugates, demonstrates that in A431 cell monolayers the membrane interaction is mostly attributable to the nanobody. For SKOV3 cells, such a clear distinction was not possible, as 7D12-r9 showed membrane staining but 7D12-pen did not. In spheroids, differences were bigger between cell lines, than between conjugates.

Altogether, the abovementioned results point towards a more prominent role of the nanobody-mediated interaction when receptor expression is high, while there is a greater influence of the CPP at lower receptor expression. Nevertheless, the CPP retains the capacity to induce endosomal uptake. 

In vitro, penetration of the tracers was better in the SKOV3 spheroids than in A431 spheroids, and both conjugates performed better than unconjugated 7D12. In vivo, uptake was higher in A431 tumors, and unconjugated 7D12 showed the highest accumulation. This discrepancy might be explained by the overexpression of EGFR in A431 cells. The high fluorescence intensity at the rims of the A431 spheroids suggests that even if penetration is incomplete, which we have not investigated in vivo, the presence of high numbers of receptors at the area that is reached still causes a high overall accumulation. A possible deeper penetration in SKOV3 tumors does not compensate for the lower amount of receptor, in terms of tracer accumulation. Relative tracer uptake did not depend on tumor size (Appendix A). Smaller tumors did not accumulate more tracer than bigger ones, which would be expected if bigger tumors showed incomplete tracer penetration or a necrotic core. Furthermore, liver sequestration of the conjugates points towards dominance of CPP-driven tissue distribution in the in vivo context. 

For both conjugates we observed increased liver and spleen uptake, and a reduction of kidney uptake. However, contrary to the observations made for an anti-HER2 affibody [44], the C-terminal His-tag present in unconjugated 7D12 did not lead to liver uptake. Interestingly, the kidney uptake of all three tracers was slightly reduced by an excess unlabeled nanobody. Labeled and unlabeled nanobody could be competing for active reabsorption at the proximal tubule, which has been observed by others for 7D12 [45].

Tumor visualization by SPECT/CT was only possible with 7D12 in A431 tumors, where tracer uptake was close to 10%ID/g. Tumors with lower uptake could not be visualized with the same settings. Others have been able to visualize tumors with only 5%ID/g nanobody uptake, but the kidney signal in the images presented appears out of scale [46]. In a clinical setting, this could pose problems in the visualization of lesions that are close to these organs. Optimization of timepoint and dose and blockade of unspecific effects by coadministration of positive amino acids or peptides could help reduce kidney accumulation of the tracer [47]. Nevertheless, high uptake in the liver would remain problematic due to high background, the possibility that it acts as a sink, and the increase in radiation exposure of the organ itself.

To our knowledge, this is first stepwise exploration of nanobody-CPP conjugates from in vitro to in vivo. So far, nanobody-CPP conjugates have been studied as cell biology tools [48] or as intracellular agents to block EGFR activation [49]. We have confirmed that nanobody-CPP conjugates can improve cell binding and internalization in vitro, both in 2D and 3D, and at different concentration ranges. However, in vivo, both tested CPPs showed a surprising extent of dominance over the functionality of the conjugate, redirecting it to the liver and counteracting a potential benefit of the CPP on tumor accumulation. Liver accumulation tends to be high for unconjugated CPPs [36] and no major changes were achieved by incorporation of targeting ligands [35]. However, a single-chain antibody conjugated to penetratin and Tat did not show increased liver targeting after 8 or 24 h, and showed moderately higher tumor retention [50]. This indicates that when the contribution of the CPP and that of the ligand are in the right balance, CPP conjugates do have the capacity to improve targeting.

Evidently, the interplay between nanobody, CPP and the cellular environment is an intricate balance that we are just beginning to understand. Therefore, more systematic comparisons of CPP targeting conjugates, as has been done for CPPs in other contexts [51,52,53], are needed to get a deeper understanding. The use of well-characterized ligands, like nanobodies, provides an excellent opportunity to compare more than one conjugate to an established “model” (e.g., the unconjugated nanobody). Advances like activatable CPPs could be useful in avoiding off-target accumulation [54,55,56], but will need to be evaluated as conjugates in the in vivo setting. In conclusion, CPPs can modify the interactions of targeting ligands with organs and tissues, while maintaining fast clearance. To exploit the full potential of CPP conjugates, the bottleneck will be to further define the structure-activity relationships of targeting agent and CPP that balance ligand- and CPP-driven interactions.

## 4. Materials and Methods

### 4.1. Nanobody Expression and Purification

The 7D12 nanobody with a periplasmic localization signal (PelB), an LPETG sortase-recognition site, an 8xHis and a VSV tag was expressed as described previously [33]. ER2566 bacteria were grown overnight in LB medium with 1% glucose and 100 µg/mL ampicillin. Then, bacterial cultures were diluted to an OD600 of 0.1 in 2xTY buffer with 5% glycerol and 50 µg/mL ampicillin and grown to log phase (OD600 = 0.6–0.8). Protein expression was induced with 0.5 mM isopropyl β-D-thiogalactoside (IPTG) at 30 °C for 3–4 h. Bacterial pellets were diluted in 0.2 M TRIS buffer, with 20% sucrose, EDTA and protease inhibitor (Roche), pH 8 and the periplasmic fraction was collected upon centrifugation. The pellet was incubated once more in the same buffer, with additional MgSO_4_, to chelate the EDTA (which could interfere with the subsequent IMAC purification). IMAC purification using NiNTA-sepharose was performed as described before [33]. Imidazole used for elution was removed by overnight dialysis in a 3.5 kDa membrane against 50 mM Tris/HCl pH 7.5, and 150 mM NaCl, or it was removed together with TCEP using a 10 kDa MWCO centrifugal unit (Amicon, Millipore, Darmstadt, Germany), after reduction of the cysteines for subsequent maleimide-dye coupling.

The control anti-GFP nanobody (sequence kindly provided by Prof. Dr. Heinrich Leonhardt, LMU München) was cloned into the same expression vector as 7D12 (periplasmic expression), and into the pQiq cytoplasmic vector. For the periplasmic expression vector, the same procedure was used as described above. For the cytoplasmic vector, expression and purification were performed as for Sortase A (see below). Both protocols yielded functional nanobody.

### 4.2. WT Sortase A Expression

Glycerol stocks of transformed ER2566 bacteria were obtained as explained previously [34] and grown overnight as described above. The overnight culture was diluted to an OD600 of 0.1 in 2xTY supplemented with 50 µg/mL ampicillin and incubated at 37 °C at 150 rpm. At OD600 ≥ 0.6, the protein expression was induced with 0.5 mM IPTG at 30 °C for 3 h. Cells were harvested by centrifugation at 3500 rpm for 20 min at 4 °C. The cell pellet was then resuspended in 50 mM Tris-HCl, pH 7.5 and 150 mM NaCl and lysed by sonication at 4 °C. The sonicated lysate was centrifuged at 6000 rpm for 15 min at 4 °C. Sortase A contained a His-tag and was purified from the supernatant via IMAC, following the same protocol as for the nanobodies. 

### 4.3. Dye and Chelator Coupling to the Nanobody 

7D12-C-LPETG-Hisx8-VSV was incubated with 20 mM TCEP for 20 min at room temperature. Excess TCEP was removed by exchanging the buffer with reaction buffer (20 mM phosphate buffer, 150 mM NaCl, 5 mM EDTA and pH 7.5) using a 10 kDa MWCO centrifugal unit (Amicon, Millipore, Darmstadt, Germany). After this step, protein concentration was measured again and maleimide Atto-532 (Sigma-Aldrich, Darmstadt, Germany) or maleimide-DTPA (Chematech, Dijon, France) was added to the solution at a protein:dye/chelator molar ratio of 1:3. This mixture was incubated for 2 h at room temperature.

The Atto532 conjugates were dialyzed overnight against a 50 mM TRIS buffer with 150 mM NaCl, pH 7.4. The DTPA conjugates were dialyzed for five days against PBS with Chelex-100 resin (Bio-Rad, Hercules, CA, USA), changing the PBS every day. After dialysis, the concentration of conjugates was measured with a Nanodrop device. For determination of protein concentrations, absorbance of the dye at 280 was corrected for and the concentrations of both protein and dye were calculated based on their absorbances and extinction coefficients. Extinction coefficients at 280 nm: 7D12: 35,535, 7D12-R9 and 7D12-r9: 34,045, 7D12-Pen and 7D12-pen: 45,045, 7D12-hLF: 39,670 and 7D12-Tat: 34,045 and at 532 nm: Atto-532: 115,000 M-1 cm^−1^.

### 4.4. Coupling of CPPs to Nanobodies Via Sortase A Transpeptidation

GGG-CPPs (R9, Penetratin, Tat, hLF, r9 and D-penetratin) where GGG stands for an N-terminal triglycyl moiety were purchased from EMC microcollections (Tübingen, Germany). Sequences can be found in Appendix A. Lower case abbreviations refer to peptides containing D-amino acids. Purified precipitates of the peptides were reconstituted in Milli-Q at a concentration of 3 mM and stored at −20 °C. 

Sortase A-mediated reactions to couple CPPs to 7D12 were performed in a maximum volume of 2 mL, with reagents at the following concentrations: 20 µM 7D12, 50 µM Sortase A and 500 µM GGG-CPP in 50 mM TRIS buffer with 150 mM NaCl and 20 mM CaCl2, pH 7.4. The mixture was incubated at 37 °C for 3 h with constant agitation at 450 rpm. Cleaved G-His8x-VSV fragments, Sortase A and unreacted nanobody were removed by reverse IMAC, as all the unwanted products contained a His-tag, while the coupled nanobodies lost it by cleavage at the LPETG Sortase A-specific site.

After reverse IMAC, unreacted GGG-CPP and calcium ions were removed by dialysis or filtration through a 10 kDa MWCO centrifugal unit, against a HEPES-buffered saline (25 mM HEPES, 0.75 mM Na2HPO4, 70 mM NaCl and pH 7). The concentration of the final products was measured as described. Conjugates were snap-frozen and stored at −80 °C.

The same procedure was performed for the anti-GFP control nanobody, with two modifications: The pI of this nanobody is predicted to be 7.09. To avoid precipitation of the nanobody, buffers were set at pH 6.5. To compensate for a lower activity of Sortase A at this lower pH, the reaction mix was incubated overnight instead of for 3 h.

### 4.5. Mass Spectrometry

The CPP conjugates were analyzed by UPLC-MS. UPLC-MS was performed on a XEVO QTOF G2 Mass Spectrometer (Waters, Herts, United Kingdom), with an Acquity H-class solvent manager, FTN-sample manager and TUV-detector. The system was equipped with a reversed phase C18-column (Waters, acquity PST 130 A, 1.7 µm 2.1 mm × 50 mm i.d.), column temperature 40 °C. Mobile phases consisted of 0.1 *v/v%* formic acid in H_2_O (buffer A) and 0.09 *v*/*v*% formic acid in 10 *v*/*v*% H_2_O in CH_3_CN (buffer B). FTN-purge solvent was 5 *v*/*v*% CH_3_CN in H_2_O. Absorption was measured at 220 nm. For the Atto532-conjugated compounds, a linear gradient of 20–80% buffer B in buffer A over 10 min was used. The DTPA-conjugates were analyzed with a linear gradient of 10–55% buffer B in buffer A over 15 min. Mass was measured in positive sensitivity mode; for a mass range between 400 and 1600 Da or 800 and 1600 Da for DTPA- or Atto532-conjugated nanobodies, respectively. Mass Spectra were deconvoluted using the MaxEnt3 software (Waters), and plotted in R. 

### 4.6. Cell Culture

All media were supplemented with 10% fetal calf serum (PAN Biotech, Aidenbach, Germany) and glutamax (Thermofisher Scientific, Waltham, MA, USA). All cell lines were incubated at 37 °C in 5% CO_2_ in a humidified atmosphere.

A431 cells were maintained in DMEM (Thermofisher Scientific) supplemented as indicated. For microscopy experiments, cells were seeded at a density of 30,000 cells per well a day before the experiment, or 15,000 cells per well two days before, in 8-well microscopy slides (Ibidi, Gräfelfing, Germany). For binding and internalization assays with radioactive compounds, A431 cells were seeded in 6-well plates at a density of 300,000 cells/well, 2 days before the experiment.

HEK293T cells (ATCC) were maintained in DMEM, supplemented as indicated. For microscopy experiments, HEK293T cells were seeded at a density of 60,000 cells per well one day before the experiment, or 30,000 cells two days before. For binding and internalization assays with radioactive compounds, HEK293T cells were seeded in 6-well plates at a density of 300,000 cells/well, 2 days before the experiment.

SKOV3 cells were maintained in DMEM (Thermofisher Scientific) supplemented as indicated. For microscopy experiments, SKOV3 cells were seeded at a density of 40,000 cells/well one day before the experiment, or 20,000 cells/well two days before in 8-well microscopy slides (Ibidi).

### 4.7. Uptake Experiments in Life Cells (Confocal Microscopy)

A431 cells were incubated for 1 h with 2 µM 7D12 and 7D12-CPP Atto532-labeled constructs, unless indicated otherwise. For the lysotracker experiments, cells were incubated with lysotracker Deep Red (Thermofisher Scientific) for 30 min at a concentration of 75 nM. All incubations were performed in phenol red-free DMEM, supplemented with 10% FCS and glutamax, and at 37 °C. After incubation cells were washed once and fresh phenol red-free medium with 20 mM HEPES was added to the wells. Cells were imaged with a laser scanning confocal microscope (LSCM) (SP5 or SP8, Leica Microsystems, Mannheim, Germany) on a temperature-controlled stage at 37 °C. We used an HCX PL APO 63 × 1.2 water immersion lens. On the SP5 system, the argon ion laser line at 514 nm was used for excitation, and emission was collected between 540 nm and 600 nm. For the SP8 system, a white light laser was used at 532 nm and emission was collected over the same range.

### 4.8. Detection of EGF Binding

For coincubation experiments with EGF, A431 cells were incubated in DMEM with 2 µM nanobody, conjugate or none (control) for 30 min at 4 °C, and then 16.5 nM EGF-biotin (Thermofisher Scientific) was added to the same media and incubated for another 30 min at 4 °C. Subsequently, cells were incubated with 2 µg/mL streptavidin-Alexa568 (Thermofisher Scientific) at 4 °C and for 30 min.

For the sequential incubation experiments, A431 cells were incubated first with 2 µM nanobody, conjugate or none for 30 min at 4 °C. Subsequently, media were removed and medium with 16.5 nM EGF-biotin was added and incubated for 30 min at 4 °C. Finally, medium was changed again to incubate cells with 2 µg/mL streptavidin-Alexa568 (Thermofisher Scientific) at 4 °C for 30 min. Cells were washed 1× with PBS and fixed with a chilled solution of 2% paraformaldehyde (PFA) for 10 min. After fixation, fresh PBS was added to the cells and LSCM imaging was performed. We used an HCX PL APO 63 × 1.2 water immersion lens, the nanobody was detected using the same parameters as stated above and EGF was imaged using a DPSS laser of 561 nm with detection of emission between 580 and 620 nm.

### 4.9. pTyr1068 Staining

Cells were incubated with 2 µM nanobody, conjugate or none in DMEM with 0.5% FCS for 1 h at 37 °C, then stimulated with 16.5 nM EGF-biotin (Thermofisher Scientific) for 10 min at 37 °C, washed 1× with PBS and fixed with 2% PFA at room temperature for 10 min. After fixation, cells were washed 1× and incubated for 20 min at RT with a 100 nM glycine solution in PBS to quench the free aldehyde groups of the PFA. After one wash with PBS, cells were permeabilized with 0.1% Triton X-100 (Sigma) in PBS for 20 min at RT.

Blocking was done for 20 min at RT with 3% BSA in PBS, and antibodies were diluted in the same buffer. Mouse anti-pTyr1068 primary antibody (clone D7A5, Cell Signaling Technology) was diluted 1:800 and incubated for 1 h at RT. Alexa647-conjugated anti-mouse secondary antibody (Life Technologies) was diluted 1:500 and incubated for 1 h. Cells were washed 1× with blocking buffer between the two incubations and after incubation with the secondary antibody. After the washing steps, PBS was added and LSCM imaging was performed. We used an HCX PL APO 63 × 1.2 water immersion lens. pTyr was imaged using an HeNe laser of 633 nm with detection of emission between 640 and 690 nm. 

### 4.10. Spheroid Culture, Incubation and Optical Clearing

Spheroids were obtained using the hanging drop method. SKOV3 and A431 cells were diluted in DMEM media with 2% *v/v* methylcellulose. On top of the lid of a petri dish, 30 µL drops (containing 15,000 cells per drop) were pipetted. The lid was flipped and placed on top of the dish bottom, which contained a small volume of PBS to prevent the spheroids from drying out. The dish with the hanging drops was incubated at 37 °C, 5% CO_2_ in a tissue culture incubator.

Incubation with 7D12 and conjugates (0.5 µM for 1 h) was performed in the drops once the spheroids had reached a sufficient size (typically after 3 days). The lid was flipped over and solutions containing the conjugate were carefully pipetted into the drop (pipetted volume did not exceed 10 µL). The lid was flipped again and placed over the bottom with PBS to incubate as usual at 37 °C. After incubation, spheroids were harvested into microcentrifuge tubes with PBS, by carefully sucking up the drops with a 1 mL pipette tip. The spheroids sunk to the bottom, PBS was removed and PBS with 4% PFA was added. The spheroids were kept in this solution, in slight agitation, for at least 40 min at RT. Once fixed, two to six spheroids of the same experimental condition were embedded in one collagen drop. Each collagen drop was placed in a well of an Ibidi 8-well microscopy slide, and the slide was kept for 40 min at 37 °C to allow collagen polymerization. Collagen embedding prevents movement of the spheroids during imaging.

Clearing of the collagen-embedded spheroids was done by incubating them in serial dilutions of fructose in Milli-Q with 0.5% *v/v* thioglycerol (which prevents the fructose solution from turning brown). This is an adaptation of the SeedB protocol [57]. Incubation was done for at least 3 h in 28.75% fructose, 5 h in 57.5% fructose and overnight in 115% fructose. Spheroids were imaged in the collagen drops immersed in the 115% fructose solution by LSCM. We used a 20× lens, and the same laser settings as for the 2D imaging of the Atto532-labeled nanobodies.

### 4.11. Radiolabeling of the 7D12 Conjugates and Quality Control

7D12-DTPA, 7D12-DTPA-Pen, 7D12-DTPA-R9 and the D-peptide conjugates were incubated with [111In]InCl3 (Curium, Petten, The Netherlands) and twice the volume of metal free 0.5 M 2-(N-morpholino)ethanesulfonic (MES) buffer, pH 5.5 for 30 min at room temperature. Labelling efficiency and radiochemical purity were determined by instant thin-layer chromatography (iTLC) on a silica gel chromatography strip (Biodex, Shirley, NY, USA), using 0.1 M citrate buffer pH 6.0 as the mobile phase. For in vitro studies, labelling was performed at 0.25 MBq/µg; for in vivo studies, labelling was performed at 2 MBq/µg for the SPECT injections (10 MBq per mouse) and at 0.2 MBq/ug for the mice, which were only used for ex vivo biodistribution (1 MBq per mouse). 

Binding and internalization assays of radiolabeled conjugates:

A431 or HEK293T cells were seeded on 6-well plates one or two days before the experiment and were used once they were confluent. Conjugates were radiolabeled following the procedure described above. Cells were incubated with 300 Bq of the conjugates in 1 mL of 0.5% BSA in RPMI (binding buffer) for 1 h at 37 °C. EGFR specificity of binding was assessed by coincubation with 50 µg unlabeled 7D12 per well. After incubation, cells were washed twice with PBS, and the receptor-bound conjugates were retrieved by incubation with ice-cold 0.1 M acetic acid, 154 mM NaCl and pH 2.6 for 10 min on ice. After washing twice with PBS, cells containing the internalized conjugates were collected with 1 mL 0.1 M NaOH. Activity in both fractions was determined in a γ-counter (2480 Wizard 3′’, LKB/Wallace, Perkin-Elmer, Waltham, MA, USA). Specific binding and internalization were calculated as a percentage of the total activity added, using solution standards.

### 4.12. Animals

Animal experiments complied with the Dutch and European regulations on animal experimentation and were performed after approval of the Animal Ethical Committee of the Radboud University Nijmegen (project number: RU-DEC-2015-0071). Female BALB/c nude mice, 6–8 weeks old, weighing approximately 20 g, were ordered from Charles River Laboratories (Lárbresle, France) and were housed at the local animal facility in groups of 6, in IVC blueline cages enriched with bedding material and one polycarbonate shelter per cage. After 1 week of acclimatization, experiments were started. After tracer injection, some mice were housed individually overnight to avoid fighting or cross-contamination through radioactive material excreted in bodily fluids. Otherwise, solitary caging was avoided.

### 4.13. In Vivo Biodistribution and SPECT Imaging

Mice were injected subcutaneously on the right flank with 0.2 mL of SKOV3 cell suspension containing 25,000,000 cells/mL (5,000,000 cells/mouse), and one week later on the left flank with 0.2 mL of a cell suspension containing A431 cells, 2,000,000 cells/mouse. When the tumors were clearly visible, mice were randomly divided into groups and the experiment was performed. 

All mice were intravenously injected with 5 µg of the corresponding radiolabeled 7D12 variant in PBS/0.5% BSA ([111In]In-DTPA-7D12 (*n*= 10), [111In]In-DTPA-7D12-r9 (*n* = 10) or [^111^In]InDTPA-7D12-pen (*n* = 10)). Four mice per group (2 per timepoint) received 10 MBq to perform SPECT imaging. The other 6 mice in each group (3 per timepoint) received 1 MBq. For the blocking groups, a 160–180-fold excess of unlabeled 7D12 was co-injected. Mice were euthanized by CO_2_/O_2_ suffocation 4 h and 24 h after injection. SPECT/CT (U-SPECT-II, MiLabs, Utrecht, The Netherlands) was performed immediately after euthanasia. The image was acquired using a 1 mm diameter pinhole ultra-high sensitivity mouse collimator. 

Blood, tumors and relevant tissues (muscle, heart, lung, spleen, pancreas, kidney, liver, stomach, duodenum and colon) of all animals were dissected, weighed and measured in a γ-counter (2480 Wizard 3″, LKB/Wallace, Perkin-Elmer). The percentage injected dose per gram tissue (% ID/g) was determined based on the cpm measured for diluted injection mixtures (standards). 

### 4.14. Data Analysis and Statistics

Microscopy images were processed with Fiji (ImageJ). For the 2D images we obtained the mean fluorescence intensity of the area covered by cells. First, a region-of-interest was created that only contained the cells. Then, an automatic threshold was applied, and a binary image was created. The total pixel intensity in the ROI was obtained and divided by the total number of positive pixels in the binary image. This was done in a semiautomated manner by using a python macro.

SPECT/CT images were reconstructed using the U-SPECT-Rec software provided by MiLabs, using an 0.4 mm voxel size, 3 iterations and 16 subsets, and visualized in Inveon Research Workplace software (version 3.0; Siemens Preclinical Solutions, München, Germany).

Files containing the results of microscopy image processing or gamma counting of the plate assays were processed in Excel, and graphs and statistical analyses were generated in GraphPad PRISM (San Diego, CA, USA). csv files containing biodistribution results were analyzed and plotted using R. The following packages were used: tidyverse, readxl and patchwork.

For the biodistribution results, organs with a weight lower than the minimal weight of the balance (0.014 g) were excluded. Outliers that could not be explained by known technical errors were kept in both the graphs and the statistical analyses. For the statistical analyses, values were log transformed and the difference in tumor uptake between the four compounds was evaluated separately for each timepoint and tumor type, with a one-way ANOVA and a Tukey’s post-hoc test. We accepted a type I alpha error of 5%. The number of animals per group was selected assuming an SD of 10% of the mean, to reach a statistical power of 80%.

## 5. Conclusions

Conjugation to cell-penetrating peptides endows targeting ligands with novel functional characteristics such as induction of cellular uptake and penetration into 3D tissues. The balance of activities depends on multiple factors among which the nature of the CPP and the level of receptor expression. The differences between CPPs are retained when going from simple monolayer cultures to more complex 3D cell cultures and in vivo models, but at the same time other functional characteristics relating to the higher level of complexity are added. Focusing future research on the factors that balance ligand- and CPP-driven interactions in complex settings, especially in vivo, will help exploit the full potential of CPP-conjugates of small ligands for targeted delivery.

## Figures and Tables

**Figure 1 pharmaceuticals-14-00602-f001:**
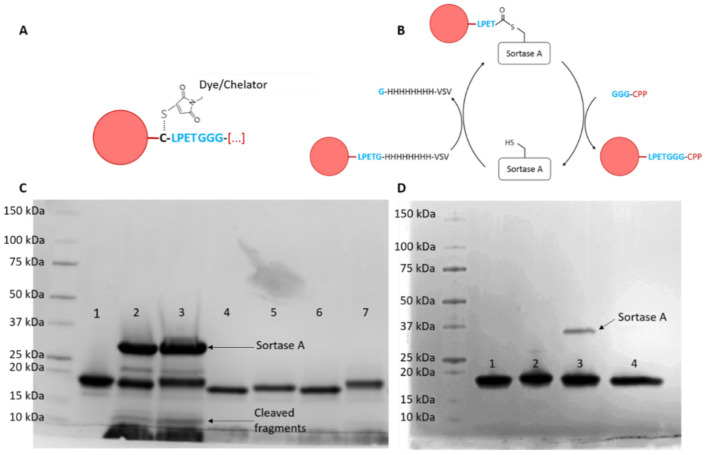
(**A**) Schematic representation of coupling of the Atto532 dye or DTPA chelator to the unpaired cysteine in 7D12 using maleimide chemistry and (**B**) sortase A-mediated transpeptidation for coupling of the various CPPs to the C-terminus of 7D12. (**C**) Coomassie brilliant blue-stained SDS-PAGE gel of 7D12-Atto532 conjugates; 1: 7D12-Atto532; 2–3: reaction mixtures: (2: 7D12-Atto532-Pen, 3: 7D12-Atto532-hLF) 4–7: purified conjugates (4: 7D12-Atto532-R9, 5: 7D12-Atto-532-Pen, 6: 7D12Atto532-Tat, 7: 7D12-Atto532-hLF). Slight differences in band height are due to different molecular weights of the peptides. (**D**) Coomassie brilliant blue-stained SDS-PAGE gel of 7D12-DTPA conjugates; 1: 7D12, 2: 7D12-DTPA, 3: 7D12-DTPA-r9, 4: 7D12-DTPA-pen. The arrow points to a small amount of Sortase A that could not be removed with purification.

**Figure 2 pharmaceuticals-14-00602-f002:**
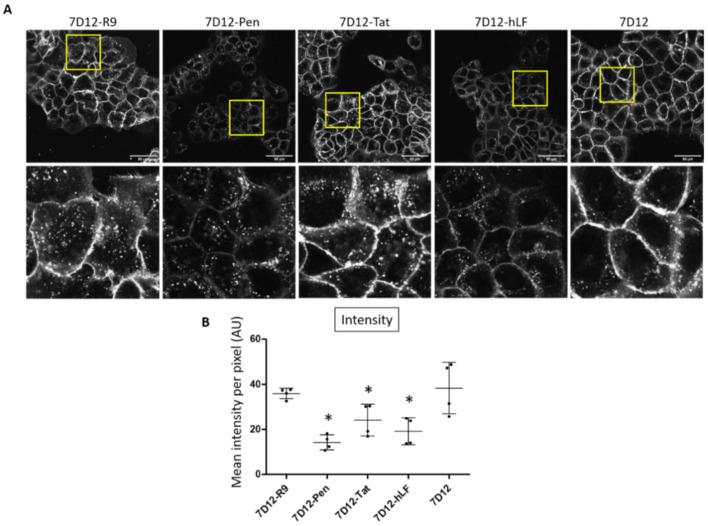
(**A**) Visual comparison of the binding and uptake of the different 7D12-Atto532-CPP conjugates in A431 cells. Upper panels: whole field of view. Lower panels: enlargements of the indicated areas. (**B**) Quantification of the mean fluorescence intensity in each field of view. Data from two independent experiments, two fields of view per experiment. Intensity values of experiment 2 were normalized to intensity values of experiment 1. However, note that for experiment 2, the overall intensity was lower. Each dot represents a field of view and error bars indicate SD. Differences in mean pixel intensity were analyzed with a one-way ANOVA and a post hoc Dunnett’s test comparing each 7D12-CPP conjugate to the unconjugated 7D12. We accepted a type I error of 5% (*p* < 0.05). Significant differences are indicated with an asterisk.

**Figure 3 pharmaceuticals-14-00602-f003:**
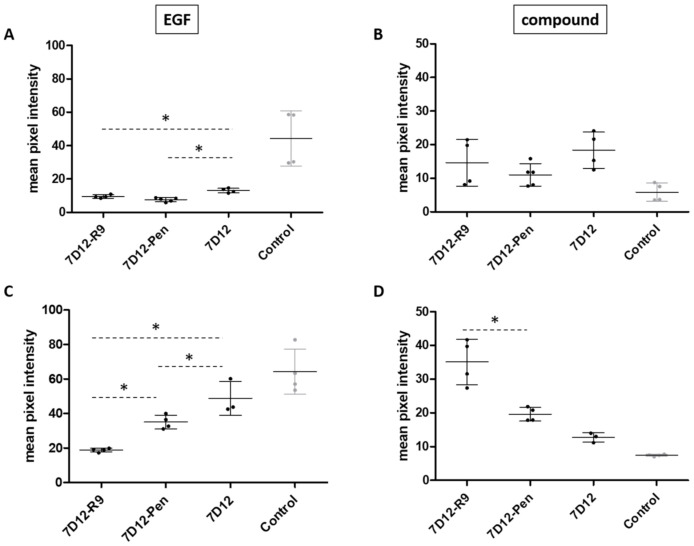
Signal intensity (arbitrary units) reflecting binding of biotinylated EGF (left) and the different conjugates (right) on A431 cells. Dots represent the mean pixel intensity of a field of view. Two fields of view were taken for each of 2 independent experiments. Error bars indicate SD. (**A**) EGF binding after coincubation. Cells were incubated at 4 °C for 30 min with nanobody, then EGF was added to the same media. (**B**) Conjugate binding after coincubation. (**C**) EGF binding after sequential incubation. Cells were incubated at 4 °C for 30 min with nanobody, then media were removed and medium with EGF was added. (**D**) Conjugate binding after sequential incubation. Differences in intensity were analyzed separately for the conjugate and for EGF, using a one-way ANOVA and a post hoc Bonferroni test comparing 7D12-R9, 7D12-Pen and 7D12 to each other. We accepted a type I error of 5% (*p* < 0.05). Significant differences are indicated with an asterisk.

**Figure 4 pharmaceuticals-14-00602-f004:**
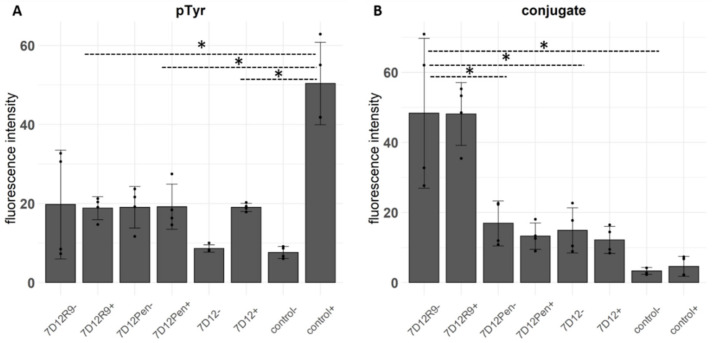
**(A**) Fluorescence intensity (arbitrary units) of immunostaining with an anti-pTyr antibody. (**B**) Fluorescence intensity of the bound conjugates. Each bar represents the mean intensity per pixel, taken from ROIs of 4 fields of view per condition, 2 from each of 2 independent experiments (±incubation without/with EGF). Each dot represents a field of view and error bars indicate SD. We tested the differences with a one-way ANOVA and a post-hoc Tukey’s test comparing all conditions to each other. We accepted a type I alpha error of 5% (*p* < 0.05). Significant differences are indicated with an asterisk.

**Figure 5 pharmaceuticals-14-00602-f005:**
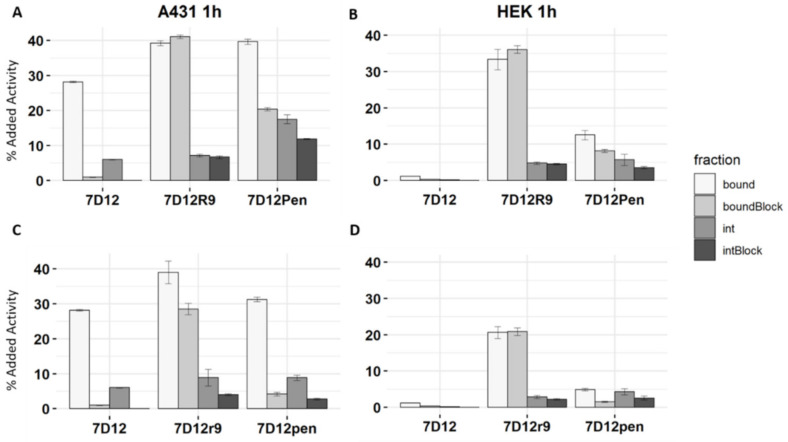
Binding and internalization assay of [111In]In-DTPA-7D12, [111In]In-DTPA-7D12-Pen and [111In]In-DTPA-7D12-R9 using A431 and HEK293 cells. (**A**) A431 cells, L-peptide conjugates (**B**) HEK cells (EGFR negative), L-peptide conjugates. (**C**) A431 cells, D-peptide conjugates. (**D**) HEK cells (EGFR negative), D-peptide conjugates. Error bars show SD of three experimental replicates.

**Figure 6 pharmaceuticals-14-00602-f006:**
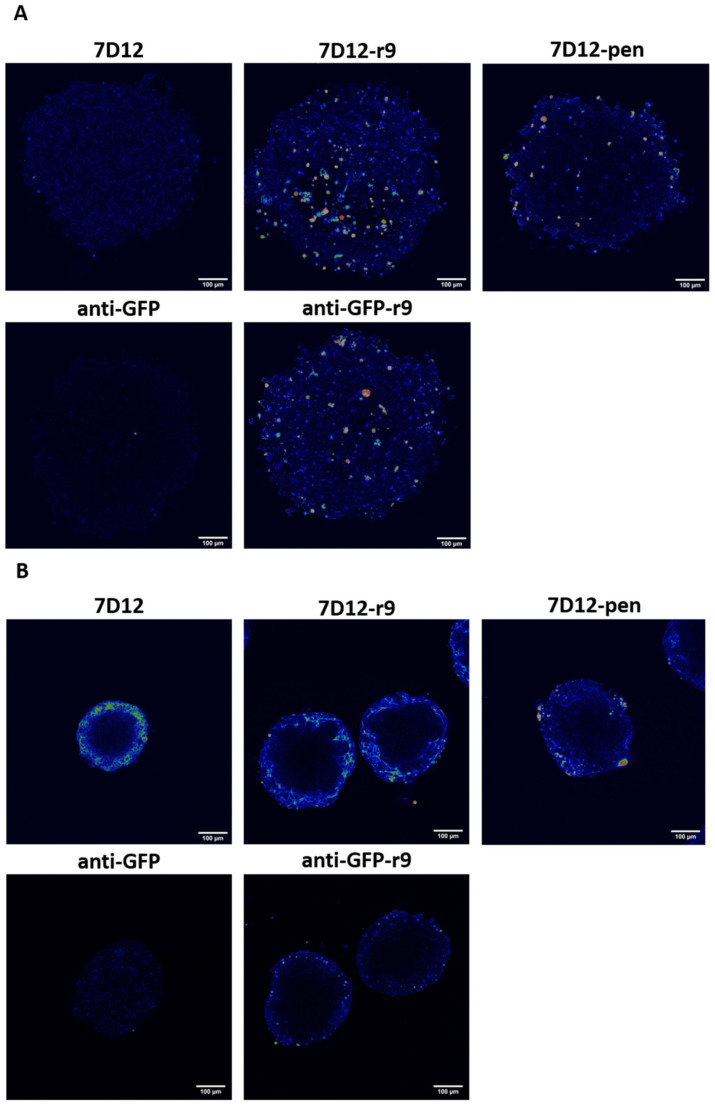
Comparison of conjugate and nanobody binding and uptake in spheroids. Incubation was done for 1 h with a concentration of 500 nM. (**A**) SKOV3 spheroids and (**B**) A431 spheroids. All spheroids were fixed and cleared to be able to visualize the whole cell mass.

**Figure 7 pharmaceuticals-14-00602-f007:**
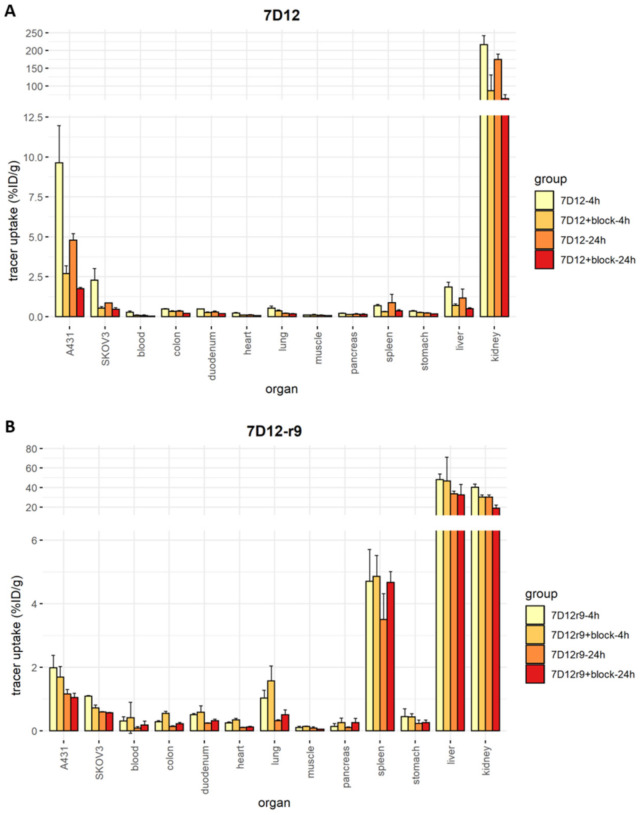
Ex vivo biodistribution results. Each dot represents one value, error bars represent SD, solid bars represent the mean. (**A**) Biodistribution of [111In]In-DTPA-7D12-r9 in all organs. (**B**) Biodistribution of [111In]In-DTPA-7D12-pen in all organs. (**C**) Biodistribution of [111In]In-DTPA-7D12 in all organs. (**D**) Tumor accumulation for each condition and cell line. Differences in accumulation between compounds were evaluated for each timepoint and tumor type with a one-way ANOVA, and a Tukey’s post-hoc test. We accepted a type I alpha error of 5% (*p* < 0.05). Significant differences are indicated with asterisks.

**Figure 8 pharmaceuticals-14-00602-f008:**
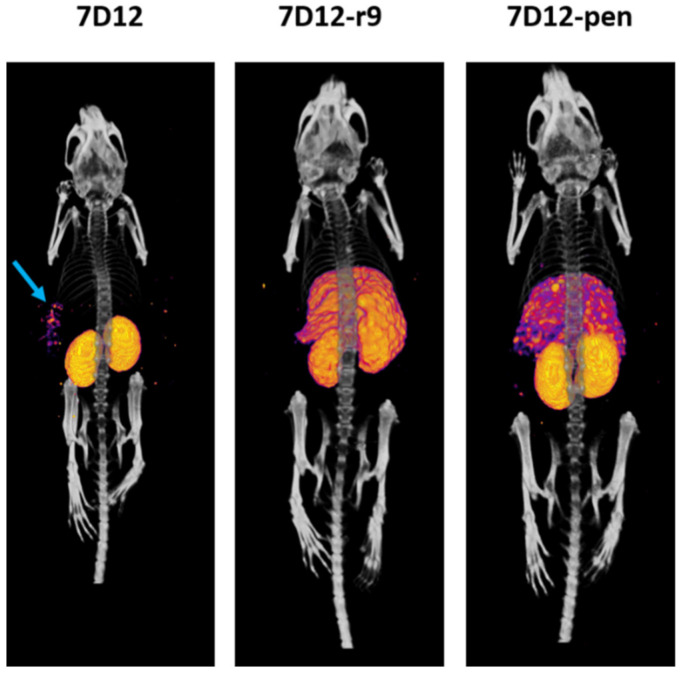
A 3D reconstruction of the SPECT/CT images of three mice. Mice were imaged post-mortem 4 h after injection of the tracer. The blue arrow points at the signal detected in the A431 tumor.

## Data Availability

Raw data of the SPECT-CT scans and microscopy images and R code used for data processing are archived at Radboudumc and available on request.

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
