# Peer review of "CPPs to the Test: Effects on Binding, Uptake and Biodistribution of a Tumor Targeting Nanobody"

_pharmaceuticals, 2021, doi:10.3390/ph14070602_

Round 1

Reviewer 1 Report

This is a very interesting study, well-thought and well-supported by data. The importance of investigating the cellular interactions in vitro and in vivo contributed by CPPs in targeting ligands will have a big impact in designing more effective localized deliveries, lessening adverse effects due to off-target accumulations. This paper maybe accepted without revision only that figure 4 formatting needs to be corrected as it covers lines 193 and 194. 

Author Response

Thank you for the time invested in reviewing our manuscript.

We have tried to address the formatting issue. We do not notice any overlap of the figure with the text, but noticed that lines are still counted next to the figure. We ask the editorial team to revise whether the figure is placed correctly during the editing process.

Reviewer 2 Report

The manuscript entitled „CPPs to the test: Effects on binding, uptake and ...“ represents a highly important study describing a comparison of nanobody functionalized with four different cell-penetrating peptides. Of these four CPPs, the study further focuses on two of the CPPs preselected based on binding and internalization in vitro. There are relatively few studies in the literature dealing with the comparison of various functionalizations (such as CPPs) or targeting moieties (such as in drug delivery systems). This is, in my opinion, this most valuable aspect of the presented manuscript. With the elevated interest in the development of theranostics, the focus on the EGFR-binding nanobodies seems to be up-to-date also in relation to potential clinical use.

The in vivo experiments belong to the undisputable advantages of the manuscript, regardless of the results rather showing better tumor accumulation and more favorable kinetics of the non-functionalized 7D12 nanobody as compared with the Pen or R9-containing conjugates.

Major comments:

  • Limited penetration of the compounds in the A431 spheroids (line 266 -270, figure 6B) is attributed to the highly dense structure of the A431 spheroids rather than the binding site barrier. Do the authors have any prove or comparison of the density of A431 and SKOV3 spheroids (such as cell number per volume or nuclei staining), in order to better explain their statement?
  • The biodistribution of the nanobody and the conjugates is a very important part of the manuscript. Therefore, presentation of the data in the Fig. 7 A, B, and C should be clearer, thus better illustrating the tracer uptake in various tissues as well as the difference between unblocked and blocked conditions. I suggest to include all the relevant numbers in the supporting results, or add detailed graphs in an appropriate scale without the values measured in liver and kidneys.
  • Do the authors have any data on the liver and kidney tracer accumulation at later time points? Also, is the tumor accumulation of the nanobodies or the conjugates dependent of the size of the tumors? In the event that such a data have been obtained, at least some discussion on this should be valuable.
  • Some data show that nonaarginin could use endocytic and non-endocytic pathways of the cell entry depending on the concentration (e.g., Melikov K. et al, Biochem. J. 2015, 471, 221-30). Could such a phenomenon be relevant also in the studied conjugates and their potential use as a theranostic agent?

Minor comments:

  • The statement in the abstract (line 23-24) should be corrected, as only two CPPs conjugated with the nanobody were studied in competition of EGF binding (as described in Section 2.3, and Fig. 3).

Inconsistency in designation of the conjugates: Pen vs pen, R9 vs r9

Author Response

Thank you for the time invested in revieweing our manusctipt, and for the useful comments.

Major questions

Question 1:

We did not perform nuclear stainings during our microscopy experiments. Therefore, we cannot provide these data. However, although we started with the same amount of cells per drop, the A431 spheroids are smaller, while A431 cells are not smaller than SKOV3 cells in 2D culture. These two observations suggest that there is a difference in the compactness of the spheroids. We rephrased the previous argument and added this evidence in the revised version.

Question 2:

We addressed this issue as well as possible by changing the scale of the graphs. The bottom part goes from 0 to 12.5 %ID/g, to include all the tumor values but facilitate the visualization of uptake in other organs other than tumors. The top part starts at 25 and ends at 200, to ensure complete visualization of liver and kidney uptake throughout the whole range of values. Furthermore, we have included supplementary tables B1 and B2, reporting the %ID/g for all organs and conditions. As our question focused on tumor accumulation, we decided to provide a separate figure where tumors can be better compared to each other. This graph has been maintained.

Question 3:

We did not measure the size of the tumors in mm3. Mice were randomized into groups when the tumor had reached a size that allowed dissection and imaging, as assessed visually. However, we made exploratory plots of tracer accumulation against tumor weight, which have been added to the supplementary figures (see revised manuscript).

Visually, there seems to be a correlation between tumor weight and absolute uptake for 7D12 and for 7D12-pen. For 7D12-r9 the differences in uptake across different weights are smaller.

It is expected that bigger organs or tissues accumulate a higher absolute percentage of tracer. This is why we correct for the weight for each organ/tissue and indicate uptake as a percentage of the injected dose per gram of tissue (%ID/g). For the corrected values, there is no correlation between weight and uptake.

Question 4:

So far, non-endocytic uptake of CPP-nanobody conjugates has only been observed in the mean micromolar range and for nanobodies conjugated to cyclic CPPs (Nat Chem. 2017 Aug;9(8):762-771.doi: 10.1038/nchem.2811). In our in vitro experiments we only observed non-endocytic uptake and in vivo, concentrations will be even lower.

Minor comments

Question 1:

This point has been addressed as indicated by the track changes function.

Question 2:

We used capital letters to refer to the L-amino acid form of the peptides, and low case letters to refer to the D-amino acid form. In experiments where the conjugates with the L form were used, we referred to the conjugates as 7D12-R9 and 7D12-Pen. In experiments where we used the D form, we referred to the conjugates as 7D12-r9 and 7D12-pen. Even if it may be confusing at a first glance, this nomenclature follows a common practice of using lower case letters for D-amino acids and allows an accurate and transparent description of our work.
